

# Intramale variation in sperm size: functional significance in a polygynous mammal

José Luis Ros-Santaella[1,2], Eliana Pintus[3] and José Julián Garde[1]

[1] SaBio, IREC (CSIC–UCLM–JCCM), Albacete, Spain
[2] Department of Animal Science and Food Processing/Faculty of Tropical AgriSciences, Czech University of Life Sciences Prague, Prague, Czech Republic
[3] Department of Veterinary Sciences/Faculty of Agrobiology, Food and Natural Resources, Czech University of Life Sciences Prague, Prague, Czech Republic

## ABSTRACT

Studies concerning the relationships between sperm size and velocity at the intraspecific level are quite limited and often yielded contradictory results across the animal kingdom. Intramale variation in sperm size may represent a meaningful factor to predict sperm velocity, due to its relationship with the level of sperm competition among related taxa. Because sperm phenotype is under post-copulatory sexual selection, we hypothesized that a reduced intramale variation in sperm size is associated with sperm competitiveness in red deer. Our results show that low variation in sperm size is strongly related to high sperm velocity and normal sperm morphology, which in turn are good predictors of male fertility in this species. Furthermore, it is well known that the red deer show high variability in testicular mass but there is limited knowledge concerning the significance of this phenomenon at intraspecific level, even though it may reveal interesting processes of sexual selection. Thereby, as a preliminary result, we found that absolute testes mass is negatively associated with intramale variation in sperm size. Our findings suggest that sperm size variation in red deer is under a strong selective force leading to increase sperm function efficiency, and reveal new insights into sexual selection mechanisms.

## INTRODUCTION

Sperm competition (competition between the sperm from two or more males for the fertilization of a given set of ova; *Parker, 1998*) is a powerful and taxonomically widespread selective force driving the evolution of sperm shape and function (*Birkhead & Møller, 1998*; *Immler, 2008*; *Pitnick, Hosken & Birkhead, 2009*; *Pizzari & Parker, 2009*; *Simmons & Fitzpatrick, 2012*). Sperm morphology is correlated with the different levels of sperm competition across the animal kingdom (for review see: *Birkhead & Immler, 2007*; *Pitnick, Hosken & Birkhead, 2009*). Across vertebrates, species with a low risk or lack of sperm competition are characterized by sperm pleiomorphism, high percentages of sperm abnormalities, and in extremes cases, sperm degenerative features (e.g., absence of the fibrous

Corresponding author
José Luis Ros-Santaella,
rossantaella@gmail.com

sheath and reduced midpiece; reviewed in: *Van der Horst & Maree, 2014*). However, when sperm competition is intense, selection is thought to favor tight quality control (*Birkhead & Immler, 2007*), for instance, by reducing the intramale variation in sperm size (e.g., insects: *Fitzpatrick & Baer, 2011*; birds: *Calhim et al., 2011*; mammals: *Šandera, Albrecht & Stopka, 2013*) and the percentage of sperm abnormalities (e.g., birds: *Rowe & Pruett-Jones, 2011*; mammals: *Gómez Montoto et al., 2011*; *Lüpold, 2013*), as well as by increasing the proportion of live sperm (e.g., insects: *García-González & Simmons, 2005*). Intraspecific studies concerning the significance of intramale variation in sperm size are quite limited but also needed, because they can reveal relationships with sperm traits associated with sperm competitiveness and contribute to a better understanding of previously reported patterns across animal populations and species (*Kleven et al., 2008*; *Laskemoen et al., 2013*).

Sperm swimming speed is probably the best-studied sperm trait relative to fertilization success (*Simmons & Fitzpatrick, 2012*; *Fitzpatrick & Lüpold, 2014*), although factors determining sperm velocity are still not well established. Sperm size has been linked to sperm velocity in a range of inter- and intraspecific studies, but these results have been inconsistent (*Humphries, Evans & Simmons, 2008*; *Kleven et al., 2009*; *Lüpold et al., 2009a*; *Fitzpatrick & Lüpold, 2014*; *Simpson et al., 2014*). For instance, in species with internal fertilization, the few intraspecific studies investigating these kinds of relationships have yielded contradictory results (e.g., mammals: *Malo et al., 2006*; *Firman & Simmons, 2010*), even within the same species (*Passer domesticus*: *Helfenstein, Podevin & Richner, 2009*; *Cramer et al., 2014*). Recently, *Mossman et al. (2013)* reported that low variation in the length of sperm components is associated with a high percentage of motile sperm in humans. Therefore, studies concerning the links between sperm kinetics and intramale variation in sperm size might provide a better understanding of sperm functionality.

It is well known that high levels of sperm competition are related to increased relative testes mass (*Birkhead & Møller, 1998*). For this reason, relative testes mass has been widely used as a proxy measure of post-copulatory sexual selection (for review see: *Simmons & Fitzpatrick, 2012*). The degree of elaboration of pre-copulatory traits also has been associated with testes size and both positive, negative or no relationships have been found (*Merilä & Sheldon, 1999*; *Preston et al., 2003*; *Malo et al., 2005a*; *Simmons & Emlen, 2006*; *Kelly, 2008*). Moreover, interspecific studies have shown that large relative testes mass is related to low variation in sperm size both between and within males, such as in birds (*Immler, Calhim & Birkhead, 2008*; *Kleven et al., 2008*) and mammals (*Šandera, Albrecht & Stopka, 2013*; *Varea-Sánchez et al., 2014*). However, intraspecific studies about the significance of the variation in testicular size are scarce even though they might reveal interesting processes of sexual selection (*Merilä & Sheldon, 1999*; *Schulte-Hostedde & Millar, 2004*; *Liao et al., 2013*).

In deer, sperm competition is likely to be more frequent in species that form large breeding groups that vary in membership from day to day (*Clutton-Brock, Guinness & Albon, 1982*), such as in the red deer (*Jarnemo, 2011*; *Pérez-González & Carranza, 2011*; *Stopher et al., 2011*). In spite of being considered a polygynous mammal, red deer calves are sometimes fathered by a non-dominant male (*Pemberton et al., 1992*), hinds show

some degree of promiscuity (*Guinness, Lincoln & Short, 1971*), and the large numbers of aggregating females should increase the mating opportunities of less competitive males because dominant males may be unable to defend large harems (*Pérez-González & Carranza, 2011*). Moreover, as a signal of risk of sperm competition in this species, the red deer have large testes in relation to their body size when compared with other species from the Cervidae family (*Clutton-Brock, Guinness & Albon, 1982*), and low within male variation in sperm size (*Malo et al., 2006*). Furthermore, *Malo et al. (2005a)* reported that relative testes size in red deer was positively related to antler size and complexity (pre-copulatory sexually selected trait), which in turn positively covaried with sperm velocity (post-copulatory sexually selected trait).

The aims of this study were twofold. First, we examined the relationships between the intramale variation in sperm size and sperm characteristics such as sperm velocity and normal morphology, which are related to male fertility in this species (*Malo et al., 2005b*; *Ramón et al., 2013*). Because sperm competition predicts intramale variation in sperm size across related taxa (see references above), low variation in sperm size in red deer may confer an advantage in post-copulatory sexual selection by enhancing sperm competitiveness. Second, taking into account that testicular size is positively related to elaborate pre-copulatory traits (antlers, male–male competition; *Malo et al., 2005a*) and shows high variability between stags (*Martínez-Pastor et al., 2005*; *Pintus, Ros-Santaella & Garde, 2015*), we hypothesized that variation in absolute testes mass is related to the intramale variation in sperm size as demonstrated at the interspecific level in several animal taxa. Under these hypotheses, we expected to find relationships between intramale variation in sperm size and sperm function, which can reveal new insights into sexual selection processes.

## MATERIALS & METHODS

### Animal sampling

This study was approved by the "Comité de Ética en Investigación de la Universidad de Castilla-La Mancha". All animals were handled in accordance with Spanish Animal Protection Regulation RD53/2013, which conforms to the European Union Regulation 2003/65/CE. Stags were legally hunted in their natural habitat in accordance with the harvest plan of the game reserve. The harvest plans were made in accordance with Spanish Harvest Regulation, Law 2/93 of Castilla-La Mancha, which conforms to the European Union regulations. Landowners and managers of the red deer populations gave permission to the authors to use the samples.

Samples were collected from 17 adult red deer (*Cervus elaphus*) males culled during the breeding season (November–December 2010) in southern Spain. Due to the fact that stags were not individually marked and they were randomly hunted from natural populations, we could not know the mating tactic employed by each male (i.e., harem-defender vs. sneaker). Testes, within the scrotal sac, were cut with a knife and transported to the laboratory in a plastic bag at ambient temperature. Samples were processed between 4 and 8 h after the death of the animals, a time lapse during which epididymal sperm characteristics do not differ significantly (*Soler & Garde, 2003*). However, because we could

not establish the exact time of death of each animal, this variable was not include as a covariate in our analyses.

## Testes mass and sperm analyses

All of the reagents were purchased from Sigma-Aldrich (Madrid, Spain), unless otherwise indicated. Testes mass was recorded to the nearest 0.01 g using an electronic balance (EK-400H balance; A&D Co., Tokyo, Japan) after removing the epididymis and the spermatic cord with a surgical blade. Though the use of relative testes mass is highly preferable for studies such as this, we were unable to quantify male body mass or body size of individual stags due to the working conditions in the field. We therefore chose to use absolute testes mass (i.e., the combined mass of the left and right testis) in the current study. Despite the limitations of such approach, absolute testes mass can still be informative, for example, as a factor that influences variation in sperm size.

Sperm morphometry was assessed as previously described (*Ros-Santaella, Domínguez-Rebolledo & Garde, 2014*). Briefly, sperm samples were directly recovered from the cauda of both epididymides and fixed in 2% glutaraldehyde-0.165 M cacodylate/HCl buffer (pH 7.3). Sperm smears were air-dried for one day, then immersed in the fixative solution for 5 min and immediately mounted, sealing the edges with DPX mountant for histology. Sperm samples were photographed using a high-resolution DXM1200 camera (Nikon, Tokyo, Japan) under phase-contrast microscopy, using an Eclipse E600 microscope (Nikon, Tokyo, Japan) and a 40X objective (Nikon, Tokyo, Japan). The resolution of the photographs was $3,840 \times 3,072$ pixels (TIFF format). A scale of 10 µm (181 pixels) was used for the measurements. The pixel size was 0.055 µm in the horizontal and vertical axes. Sperm lengths were assessed using ImageJ software (National Institutes of Health, Bethesda, MD, USA). We measured the following sperm traits: head width, head length, flagellum length, and midpiece length. From these measurements, we calculated other morphometric parameters such as total sperm length, principal plus terminal piece length, head area, head perimeter, and head ellipticity (head length/head width). The main structures of red deer spermatozoon are shown in Fig. 1. A total of 25 spermatozoa per male were measured, as previously described (*Malo et al., 2006*; *Ros-Santaella, Domínguez-Rebolledo & Garde, 2014*).

Sperm morphology was evaluated under phase-contrast microscopy using an Eclipse E600 microscope (Nikon, Tokyo, Japan), and a 40X objective (Nikon, Tokyo, Japan). Two hundred spermatozoa per animal were assessed. Normal sperm morphology was calculated as the proportion of spermatozoa without any morphological abnormalities (e.g., pyriform head, detached head, macrocephalic and microcephalic head, bent midpiece, coiled tail, and proximal cytoplasmic droplet). A single trained researcher assessed sperm morphology in order to avoid different evaluation criteria. An example of normal sperm phenotype is shown in Fig. 1.

Sperm number was calculated by multiplying the epididymal sperm concentration (assessed using a Bürker chamber) by sperm volume. Sperm kinetics was assessed by a CASA (Computer Assisted Sperm Analysis) system. Epididymal sperm were diluted

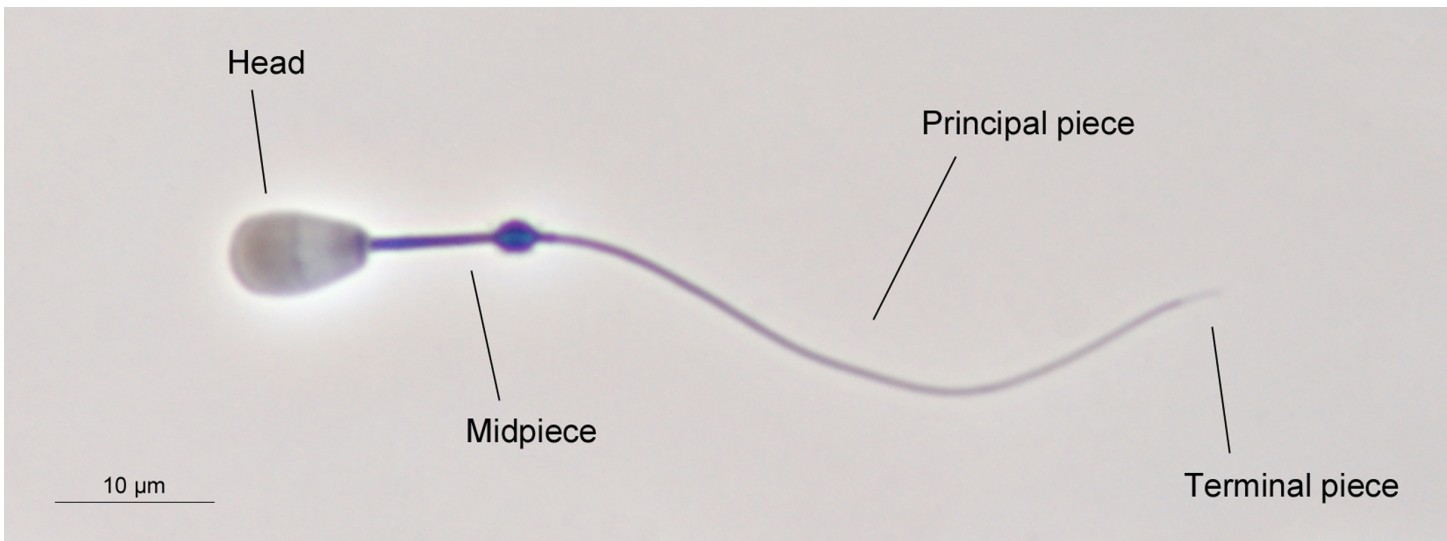

**Figure 1** **The main structures of red deer spermatozoon.** The photograph was taken under phase contrast microscopy (40X objective). Scale bar, 10 μm.

down to $25 \times 10^6$ spermatozoa/mL with bovine gamete medium (BGM-3, for medium composition see: *Domínguez-Rebolledo et al., 2010*) and incubated at 37 °C in a water bath for 15 min. After dilution and incubation, a sub-sample of 5 μL was loaded into a Makler counter chamber (chamber depth: 10 μm; Sefi-Medical instruments, Haifa, Israel) at 37 °C for kinetics analysis. The CASA system consisted of an Eclipse 80i triocular optical phase contrast microscope (Nikon, Tokyo, Japan), equipped with a warming stage at 37 °C and a Basler A302fs digital camera (Basler Vision Technologies, Ahrensburg, Germany). Videos were captured and analyzed using the Sperm Class Analyzer software (Microptic S.L., Barcelona, Spain). The analysis was carried out using a 10X negative phase-contrast objective (Nikon, Tokyo, Japan). A total of four descriptors of sperm motility were recorded analyzing a minimum of 250 sperm per sample: average path velocity (VAP, μm/s), curvilinear velocity (VCL, μm/s), straight linear velocity (VSL, μm/s), and progressive sperm motility (%). Progressive sperm motility was defined as the sperm that swim forward in a straight line (i.e., straightness ≥ 80%). Thus, spermatozoa with an abnormal swimming path (e.g., tight circles) were not included in the proportion of progressive sperm motility. Average path velocity (VAP), VCL, and VSL are strongly related to fertility in this species (*Malo et al., 2005b*). The standard parameter settings were as follows: 25 frames/s; 20–60 μm$^2$ for the head area; and a VCL > 10 μm/s to classify a spermatozoon as motile, as previously described (*Domínguez-Rebolledo et al., 2011*). Additionally to these cut-off settings, each recorded video was visually checked in order to ensure that neither debris nor dead-spermatozoa were included as motile sperm.

## Statistical analysis

All statistical analyses were performed using the SPSS 20.0 statistical software package (IBM Inc, Chicago, IL, USA). The Kolmogorov–Smirnov and Levene's tests were used

to check data normality and homogeneity of variance, respectively. The intramale coefficient of variation (CV = standard deviation/mean × 100) was calculated using the measurements of each spermatozoon, for each male. We also calculated the intermale CV from the average of each morphometric parameter. To check for differences between the intramale CV of the sperm head and flagellum components, we used paired samples Student's $t$-test. For this analysis we used the average intramale CV of the sperm head (length, width, area, ellipticity, and perimeter) and flagellum parameters (flagellum, midpiece, and principal plus terminal piece lengths) for each male. Absolute testes mass was $\log_{10}$ transformed due to the high variability found between males. Principal component analysis (PCA) was performed to obtain a single parameter describing sperm velocity. Average path velocity (VAP) was significantly correlated with VSL and VCL ($r = 0.86$; $P < 0.001$ and $r = 0.54$; $P = 0.024$, respectively), whereas, VCL and VSL were positively associated, but not significantly ($r = 0.10$; $P = 0.691$). Principal components with an eigenvalue >1 were retained. Pearson's correlation (two-tailed) test was used to assess the relationships between the intramale CV for the sperm size parameters with absolute testes mass and the different sperm traits evaluated (i.e., sperm velocity, progressive sperm motility, normal sperm morphology, and sperm number).

## RESULTS

### CV in sperm size and its relationships with sperm kinetics and normal sperm morphology

The descriptive statistics of sperm parameters are shown in Table 1. The mean value of the intramale CV of the total sperm length was low (1.31%). Similarly, mean values of the remaining intramale CV of the sperm morphometry parameters were generally low to moderate, ranging from 1.33% to 4.56% (Table 1 and Fig. 2A). Moreover, the intramale CV of the flagellum measures was significantly lower than that found in the sperm head measures ($t = 17.53$; $df = 16$; $P < 0.001$; Fig. 2B). The intermale CV of the sperm head and the flagellum parameters were also low to moderate, ranging from 1.56% to 3.62% (Fig. 2C). Principal component analysis (PCA) rendered only one component with an eigenvalue >1 that explained 68.77% of the variance in the sperm velocity variables (Table 2). All sperm velocity variables (VAP, VCL, and VSL) were positively and significantly related to PC1 scores ($P < 0.05$; Table 2). The eigenvalues and the variance explained by the other components are also shown in Table 2. Sperm velocity showed negative relationships with the intramale CV of the total sperm length ($r = -0.54$; $P = 0.025$; Fig. 3A and Table S1), head ellipticity ($r = -0.49$; $P = 0.047$; Fig. 3B and Table S1), flagellum length ($r = -0.61$; $P = 0.010$; Fig. 3C and Table S1), and principal plus terminal piece length ($r = -0.61$; $P = 0.009$; Fig. 3D and Table S1). The percentage of sperm with progressive motility showed negative relationships with the intramale CV of the total sperm length ($r = -0.49$; $P = 0.045$; Fig. 4A and Table S1), flagellum length ($r = -0.57$; $P = 0.016$; Fig. 4B and Table S1), principal plus terminal piece length ($r = -0.63$; $P = 0.007$; Fig. 4C and Table S1), and head ellipticity ($r = -0.63$; $P = 0.007$; Fig. 4D and Table S1). Moreover, the intramale CV of the sperm head area

**Table 1 Sperm parameters in red deer ($N = 17$).**

| Assessed parameters | Mean ± SD | Range (min-max) |
|---|---|---|
| *Sperm kinetics, morphology, and sperm number* | | |
| VAP (µm/s) | 104.24 ± 9.95 | 86.73–119.71 |
| VCL (µm/s) | 154.41 ± 14.60 | 129.19–179.14 |
| VSL (µm/s) | 73.03 ± 11.81 | 53.45–93.53 |
| Progressive motility (%) | 34.43 ± 10.41 | 19.11–55.93 |
| Normal morphology (%) | 86.50 ± 7.00 | 74.00–97.00 |
| Sperm number ($10^6$) | 1,208.51 ± 473.34 | 278.83–1,889.47 |
| *Intramale CV in sperm morphometry (%)* | | |
| Head width | 3.37 ± 0.54 | 2.48–4.65 |
| Head length | 2.92 ± 0.51 | 2.12–3.99 |
| Head area | 4.56 ± 0.78 | 3.36–5.80 |
| Head perimeter | 2.31 ± 0.44 | 1.56–3.06 |
| Head ellipticity (length/width) | 4.34 ± 0.69 | 3.45–6.21 |
| Sperm length | 1.31 ± 0.38 | 0.88–2.16 |
| Flagellum length | 1.33 ± 0.41 | 0.91–2.38 |
| Midpiece length | 2.47 ± 0.53 | 1.92–3.74 |
| Principal plus terminal piece length | 1.76 ± 0.50 | 1.26–2.91 |

**Notes.**

VAP, average path velocity; VCL, curvilinear velocity; VSL, straight linear velocity; SD, standard deviation; CV, coefficient of variation.

**Table 2 Principal component analysis (PCA) to determine overall sperm velocity.**

| Sperm velocity variables | PC1 loadings | |
|---|---|---|
| | Eigenvectors | *P* |
| VAP | 0.991 | <0.0001 |
| VCL | 0.590 | 0.0127 |
| VSL | 0.856 | <0.0001 |
| Variance explained (%) | 68.767 | |
| Eigenvalue | 2.063 | |

**Notes.**

Principal components with an eigenvalue >1 are shown. PC1 loadings (i.e., eigenvectors and *P* values) are correlation coefficients between the PC1 scores and the original variables of sperm velocity (VAP, VCL, and VSL). The results for the PC2 and PC3 were: PC2 (eigenvalue: 0.906; variance explained: 30.204%) and PC3 (eigenvalue: 0.031; variance explained: 1.029%).

VAP, average path velocity; VCL, curvilinear velocity; VSL, straight linear velocity; PC, principal component.

and perimeter were negatively related to the percentage of normal sperm morphology ($r = -0.54$; $P = 0.027$ and $r = -0.53$; $P = 0.029$, respectively; Table S1). The Pearson's correlations of the intramale CV in the sperm morphometry parameters with sperm kinetics and normal sperm morphology are shown in Table S1.

## Absolute testes mass, CV in sperm size, and sperm number

As expected, the absolute testes mass exhibited high variability between males (CV = 43.38%) ranging from 47.49 g to 165.51 g with an average mass of 88.55 ± 38.42 g

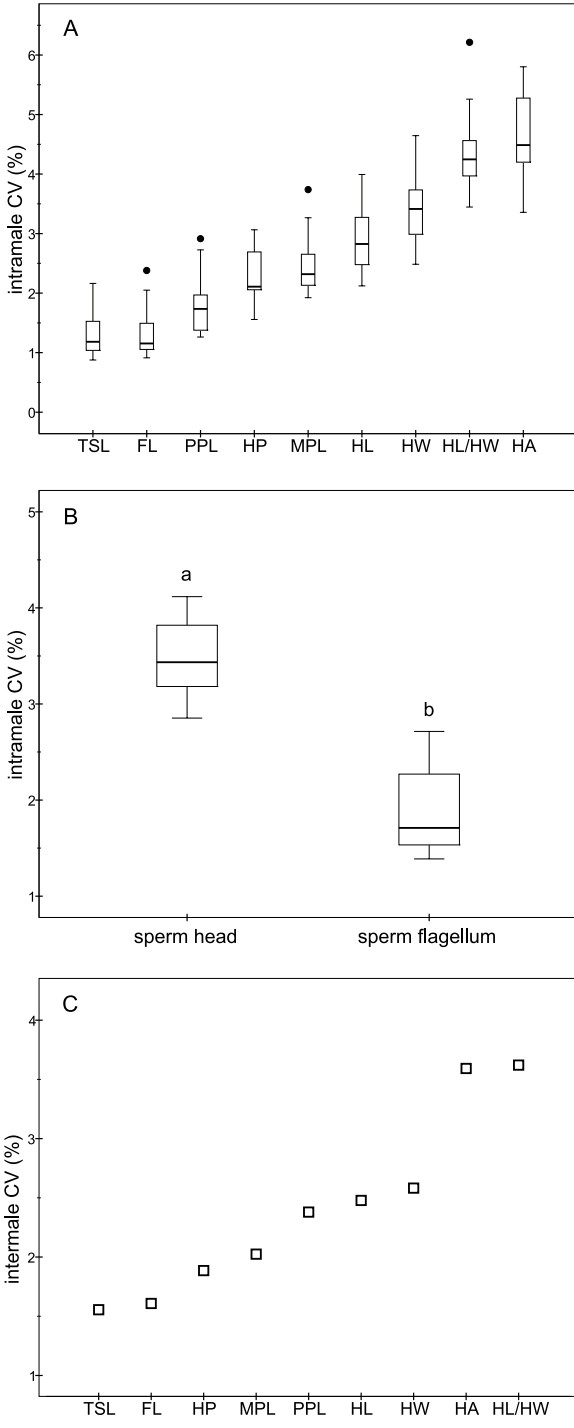

**Figure 2 Coefficients of variation in sperm morphometry parameters.** (A) Intramale coefficient of variation of the sperm morphometry parameters in red deer stags ($N = 17$); (B) Differences between the intramale coefficient of variation of the sperm head and flagellum components. Different letters differ significantly ($P < 0.001$); (C) Intermale coefficient of variation in sperm size. CV, coefficient of variation; TSL, total sperm length; FL, flagellum length; PPL, principal plus terminal piece length; HP, head perimeter; MPL, midpiece length; HL, head length; HW, head width; HL/HW, head ellipticity; HA, head area.
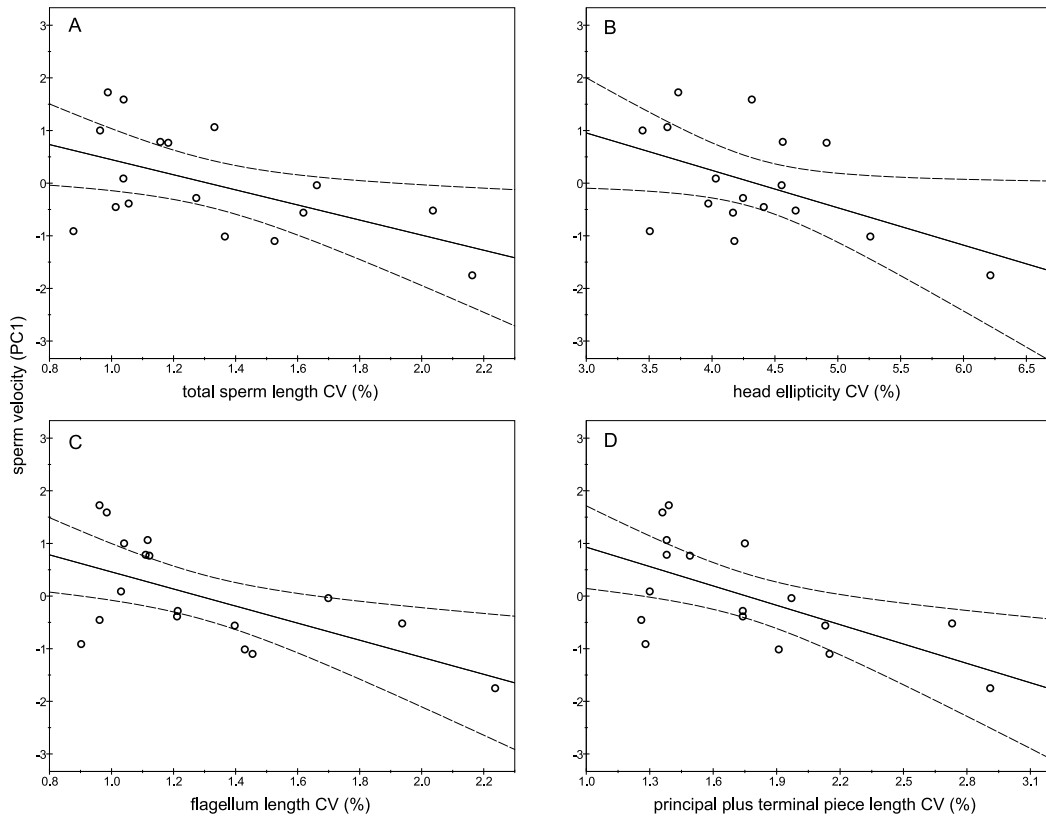

**Figure 3 Relationships between sperm velocity and the intramale coefficient of variation in sperm size.** Relationships between sperm velocity and the intramale CV of: (A) total sperm length: $r = -0.54$; $P = 0.025$; (B) head ellipticity: $r = -0.49$; $P = 0.047$; (C) flagellum length: $r = -0.61$; $P = 0.010$; (D) principal plus terminal piece length: $r = -0.61$; $P = 0.009$. CV, coefficient of variation; PC1, principal component 1.

(Mean ± SD). We found significant and negative relationships between absolute testes mass and the intramale CV of total sperm length ($r = -0.58$; $P = 0.016$; Fig. 5A and Table S1), sperm head width ($r = -0.70$; $P = 0.002$; Fig. 5B and Table S1), and midpiece length ($r = -0.49$; $P = 0.046$; Fig. 5C and Table S1). The same relationship trends were also found for the remaining intramale CV of the sperm morphometry parameters, although they were not significant ($P > 0.05$; Table S1). We did not find any significant relationship between the intramale CV in sperm size and sperm number ($P > 0.05$; Table S1). Finally, we found a positive relationship between absolute testes mass and sperm number, but it was not significant ($r = 0.35$; $P = 0.371$).

## DISCUSSION

In the present work we provide empirical evidence that intramale variation in sperm size is significantly associated with sperm velocity in a polygynous mammal, the red deer. Sperm with low variation in their size swim faster. To our knowledge, this relationship between sperm size variation and sperm velocity has never been described before in mammals and can significantly contribute to the understanding of related patterns and processes

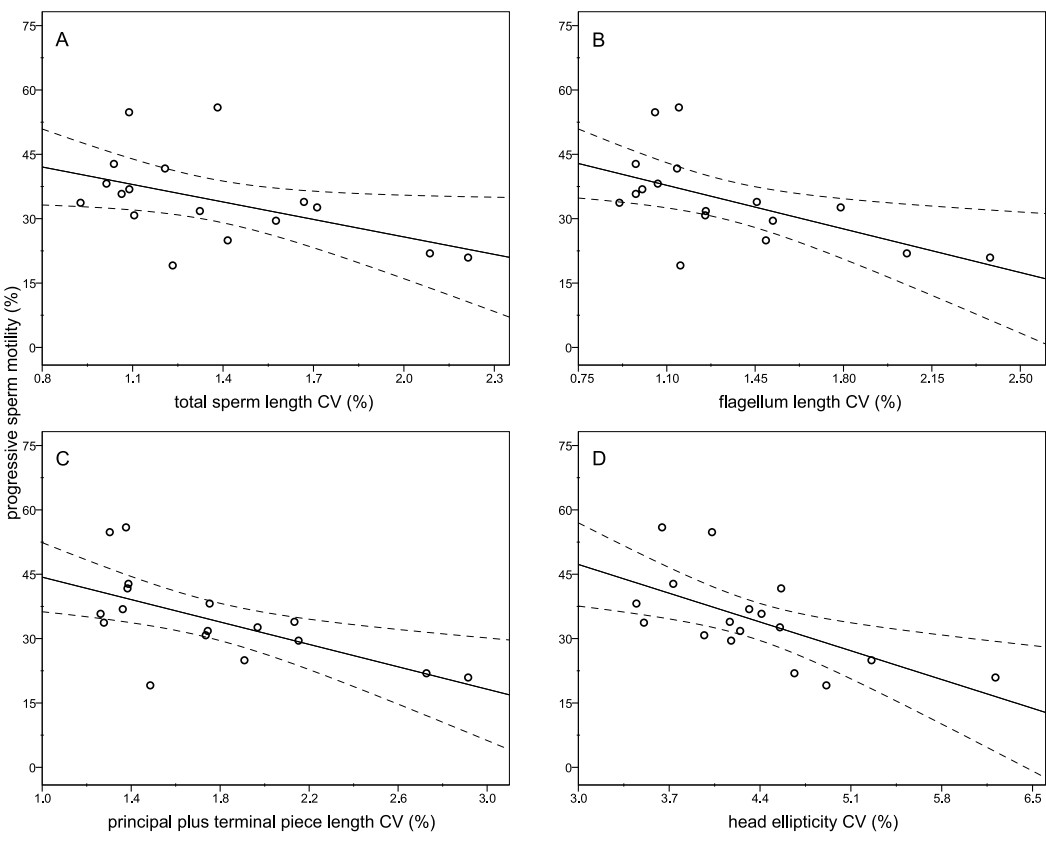

**Figure 4** **Relationships between the percentage of sperm with progressive motility and the intramale coefficient of variation in sperm size.** Relationships between progressive sperm motility and the intramale CV of: (A) total sperm length: $r = -0.49$; $P = 0.045$; (B) flagellum length: $r = -0.57$; $P = 0.016$; (C) principal plus terminal piece length: $r = -0.63$; $P = 0.007$; (D) head ellipticity: $r = -0.63$; $P = 0.007$. CV, coefficient of variation.

observed across animal populations and species (*Immler, Calhim & Birkhead, 2008*; *Laskemoen et al., 2013*). In addition, intramale variation in sperm head size is negatively related to normal sperm morphology. Highlighting the significance of our findings, previous studies reported that sperm velocity and normal morphology are related to male fertility and the sex ratio at the time of birth in red deer (*Malo et al., 2005b*; *Gomendio et al., 2006*), suggesting that post-copulatory sexual selection reduces variation in sperm size. Moreover, absolute testes mass was negatively related to intramale variation in sperm size (i.e., sperm length, head width, and midpiece length), though we suggest this result is treated with caution as both relative testes mass and male mating tactic (i.e., harem defender versus sneaker) could influence this relationship. This trend of relationship between testes mass and intramale CV in sperm size is linked to the level of sperm competition and extrapair paternity among related taxa (*Kleven et al., 2008*; *Šandera, Albrecht & Stopka, 2013*). Our results tentatively suggest that large testes in red deer lead to sperm closer to the putative optimal size, which potentially enhances sperm competitiveness.

The consequences of having a low variation in sperm size still remain poorly investigated, and might be expected to improve some sperm traits related to fertilization

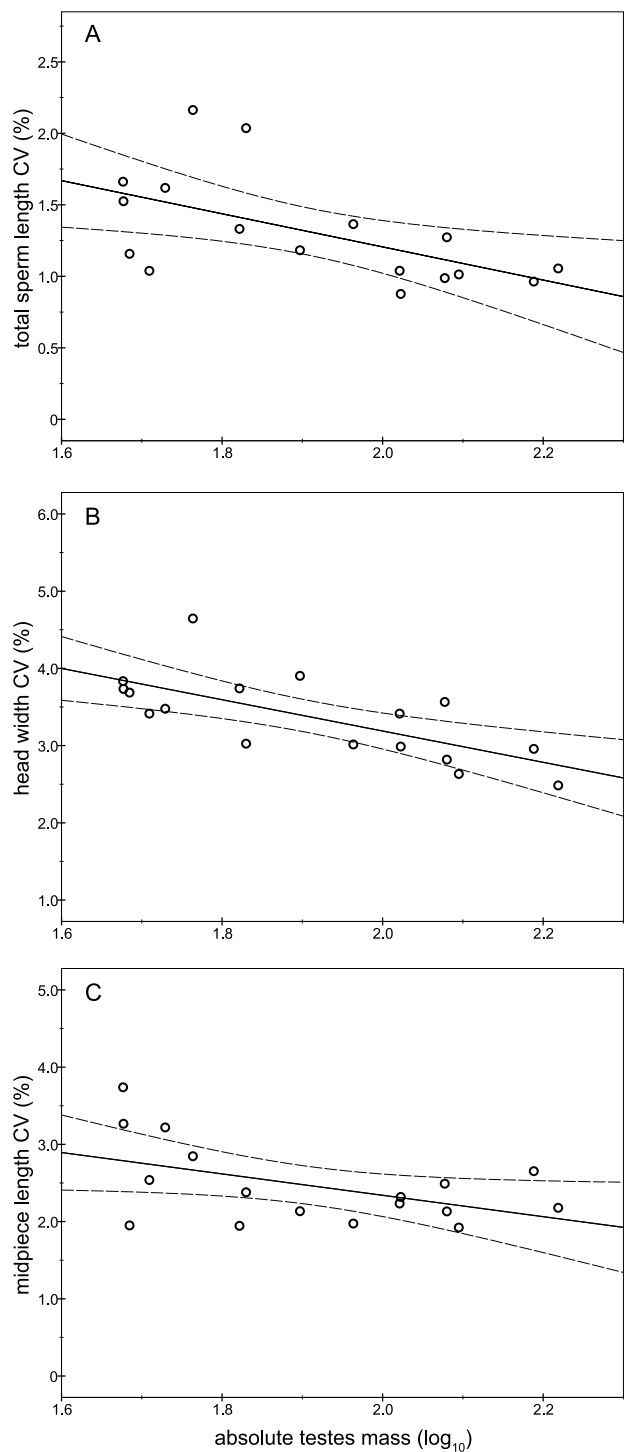

**Figure 5 Relationships between absolute testes mass (log$_{10}$) and the intramale coefficient of variation in sperm size.** Relationships between absolute testes mass and intramale CV of: (A) total sperm length: $r = -0.58$; $P = 0.016$; (B) head width: $r = -0.70$; $P = 0.002$; (C) midpiece length: $r = -0.49$; $P = 0.046$. CV, coefficient of variation.

success. As a generalization of our study, intramale and intermale variations in sperm size were low both for the head and the flagellum structures, suggesting that in red deer stabilizing selection might act on sperm to reduce their variation in size. Moreover, we found that the variation in sperm measures was higher for those of the sperm head than for the sperm flagellum, as previously reported in murine rodents (*Varea-Sánchez et al., 2014*). Such a result may be linked to the size difference between X and Y chromosome bearing spermatozoa found in humans (X is larger than Y; *Cui & Matthews, 1993*), as these size differences are more significant for the sperm head than for the flagellum components (*Cui & Matthews, 1993*; *Cui, 1997*). Recently, *Mossman et al. (2013)* reported that the variation in the length of sperm components in humans undergoing treatment for infertility correlated with the proportion of motile sperm, but the authors did not find any association with sperm velocity parameters. However, in two closely related species of lizards, the relationships between the intramale variation in sperm size and sperm velocity showed contradictory results (*Blengini et al., 2014*). In agreement with *Mossman et al. (2013)*, we found that the intramale variation in sperm size is negatively related to the percentage of sperm with progressive motility. Further, in our study, we found a strong negative relationship between the intramale variation in sperm size and sperm velocity, which in turn is highly related to male fertility in red deer (*Malo et al., 2005b*).

In the present study, the intramale CV of the principal plus the terminal piece length showed a strong relationship with sperm velocity. On the contrary, we did not find any significant relationship between sperm velocity and the intramale CV of the sperm midpiece length, probably because glycolysis, which appears to be carried out along the length of the principal piece, is the most important source of ATP for the tail, instead of oxidative phosphorylation in the midpiece (*Turner, 2006*). Further, in red deer, the length of the principal plus the terminal piece in relation to the rest of the flagellum as well as head ellipticity play an important role in determining sperm velocity (*Malo et al., 2006*). For this reason, ejaculates with more uniform size of these sperm traits may be more successful at achieving fertilizations under competitive mating scenarios. Moreover, we found that the intramale variation in the sperm head size (area and perimeter) is negatively correlated with the percentage of normal sperm morphology, which in turn plays a key role in male fertility as well as in the offspring's sex ratio in red deer (*Malo et al., 2005b*; *Gomendio et al., 2006*). In the present study, a reduced intramale variation in sperm size was related to high sperm velocity and normal morphology, which might be indicative of an efficient spermatogenesis. On the other hand, the factors and mechanisms involved in the production of sperm with low variation in their size still remain unknown and deserve further investigation. It is well known that Sertoli cells are involved in several important processes, such as the elongation of spermatids and the formation of sperm tails (*Griswold, 2004*), among others. Because Sertoli cell number covaries with testis mass and sperm function in red deer (*Pintus, Ros-Santaella & Garde, 2015*), we hypothesize that Sertoli cells can also play a role in determining the intramale variation in sperm size, in agreement with the hypothesis given by *Mossman et al. (2013)*. A high Sertoli cell number or functionality may lead to better-manufactured sperm, thus enhancing sperm function.

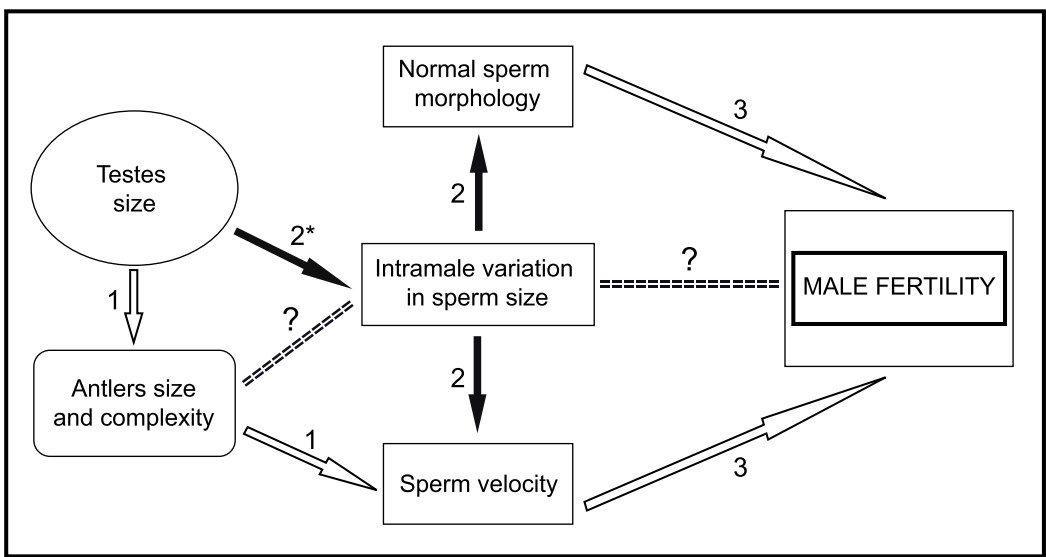

**Figure 6 General scheme of male traits related to sperm competitiveness and fertility in red deer.** White arrows indicate positive relationships. Black arrows indicate negative relationships. (1) *Malo et al., 2005a*; (2) Present work; (3) *Malo et al., 2005b*. \*, In the present work we have used absolute testes mass instead of relative testes mass. ?, Needs to be evaluated.

We found high variability in absolute testes mass among red deer males as previously described (*Martínez-Pastor et al., 2005*; *Pintus, Ros-Santaella & Garde, 2015*). Studies in natural populations of ungulates, such as red deer and Soay sheep, have shown that testes size (both relative and absolute) is positively correlated with antler and horn size (*Preston et al., 2003*; *Malo et al., 2005a*), copula rates, and siring success (*Preston et al., 2003*). Our results show that large absolute testes mass is also associated with low sperm size variability. Furthermore, we found a positive relationship between absolute testes mass and sperm number, but it was not significant. We speculate that these results may reflect an increase in testes mass in order to produce more size-uniform sperm as occurs at intraspecific level (birds: *Immler, Calhim & Birkhead, 2008*; mammals: *Šandera, Albrecht & Stopka, 2013*). This idea is somewhat supported by the fact that, across species, both sperm morphology and sperm number are influenced by testicular architecture (e.g., relative proportion of spermatogenic tissue and size of the tubule lumen; *Lüpold et al., 2009b*; *Rowe & Pruett-Jones, 2011*) and the efficiency of spermatogenesis (e.g., shorter duration of the cycle of the seminiferous epithelium; *Ramm & Stockley, 2010*).

In conclusion, in the present work, we provide empirical evidence that low variation in sperm size is associated with high sperm velocity and normal morphology. Taking into account that these last two sperm traits are good predictors of male fertility in red deer (*Malo et al., 2005b*), our findings suggest that sperm size variation is under strong post-copulatory sexual selection in this species. Furthermore, we found that absolute testes mass is negatively associated with intramale variation in sperm size. However, while our results are robust, they need to be considered with some caution because of the sample size used. Moreover, male mating tactics and relative testes mass were unknown

and could have an effect on the main sperm traits evaluated. With the findings of the present work, we can add new information to the general scheme about the male traits that potentially enhance sperm competitiveness, which in turn determine male fertility in red deer (Fig. 6). Our next step is to examine the possible relationships between sperm phenotype, testicular architecture, and spermatogenic features, as well as whether intramale variation in sperm size is a predictor of male fertilization ability. Our results highlight the importance of assessing intramale variation in sperm size for an enhanced understanding of post-copulatory sexual selection mechanisms.

## ACKNOWLEDGEMENTS

The assistance provided by Alvaro Domínguez-Rebolledo, Alfonso Bisbal, Enrique del Olmo, Mari Cruz Sotos, and Zandra Maulen is gratefully acknowledged. Authors acknowledge Gregorio Moreno-Rueda for his insightful comments and Christina Baker for English language editing. Landowners, managers, and rangers are acknowledged for facilitating access to samples.

### Funding

JLRS and EP were supported by the project CIGA 20145001 (Czech University of Life Sciences, Prague, Czech Republic). The funders had no role in study design, data collection and analysis, decision to publish, or preparation of the manuscript.

### Grant Disclosures

The following grant information was disclosed by the authors:
Czech University of Life Sciences: CIGA 20145001.

### Competing Interests

The authors declare there are no competing interests.

### Author Contributions

- José Luis Ros-Santaella conceived and designed the experiments, performed the experiments, analyzed the data, wrote the paper, prepared figures and/or tables, reviewed drafts of the paper.
- Eliana Pintus conceived and designed the experiments, performed the experiments, analyzed the data, wrote the paper, reviewed drafts of the paper.
- José Julián Garde conceived and designed the experiments, contributed reagents/materials/analysis tools, reviewed drafts of the paper.

### Animal Ethics

The following information was supplied relating to ethical approvals (i.e., approving body and any reference numbers):

This study was approved by the "Comité de Ética en Investigación de la Universidad de Castilla-La Mancha." All animals were handled in accordance with Spanish Animal

Protection Regulation RD53/2013, which conforms to the European Union Regulation 2003/65/CE. Stags were legally hunted in their natural habitat in accordance with the harvest plan of the game reserve. The harvest plans were made in accordance with Spanish Harvest Regulation, Law 2/93 of Castilla-La Mancha, which conforms to the European Union regulations. Landowners and managers of the red deer populations gave permission to the authors to use the samples.

## Data Availability

The data set is included as Data S1.

## Supplemental Information

Supplemental information for this article can be found online at http://dx.doi.org/10.7717/peerj.1478#supplemental-information.

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
