# Peer review of "Intramale variation in sperm size: functional significance in a polygynous mammal"

_PeerJ, doi:10.7717/peerj.1478_

## Round 0.1 · original submission · Major Revisions

Please considered all the suggestions in any revised version of the manuscript. In particular, the comments of Reviewer 2 should be carefully addressed.

Reviewer 1 ·

Basic reporting

The degree of postcopulatory sexual selection, comprising variable degrees of sperm competition and cryptic female choice, is an important evolutionary force to influence sperm form and function. The authors of this paper examine intramale variation in sperm size: functional significance in a polygynous mammal. They find that high testes mass is associated with low variation in sperm size. Moreover, low variation in sperm size is strongly related to high sperm velocity and normal sperm morphology, which in turn are good predictors of male fertility in this species. Based on these results, the authors conclude that sperm size in red deer is under a strong selective force and reveal new insights into sexual selection mechanism. Overall, I enjoyed the manuscript, and think the overall data set is interesting, as is the evidence for species had distinct testicular morphological characteristics, reflecting sperm competition. The paper is generally well written: the introduction sets the scene well, methods are in general appropriate and well described, and the conclusions in the discussion are mostly well supported by the data. However, I also find that data is small in the paper. However, in the introduction for author in PeerJ, PeerJ evaluates articles based only on an objective determination of scientific and methodological soundness, not on subjective determinations of 'impact,' 'novelty' or 'interest'. Hence, I think that the paper can be published in PeerJ with a minor revision.
Authors should notice two minor comments: Firstly, authors should read trecent global reference about sperm competition within or among species in frogs. For example, recent references was in also referred sperm size in anurans by Zeng et al. (2014) and intramale variation in Mi et al. (2012) and Liao et al. (2013).
Zeng, Y., Lou, S.L., Liao W.B., Jehle, R. 2014. Evolution of sperm morphology in anurans: insights into the roles of mating system and spawning locations. BMC Evolutionary Biology, 14: 104
Liao, W.B., Mi, Z.P., Li, C.L., Wei, S.C., Wu, H. 2013. Sperm traits in relation to male amplexus position in the Omei treefrog Rhacophorus omeimontis, a species with group spawning. Herpetological Journal, 23: 23-27
Mi, Z.P., Liao, W.B.*, Jin, L., Lou, S.L., Cheng, J., Wu, H. 2012. Testes asymmetry and sperm length in Rhacophorus omeimontis. Zoological Science, 29: 368-372
Second, authors should discuss the sperm competition in frog associated with the other frog species. Finally, authors should check carefully references. See attached file.

Experimental design

Measurements comprised 20 sperms from each male need be done. All measurements (testes and sperm) were taken without knowledge of the species identification to prevent observer bias. The repeatability was high or low when we compared three measurements on 20 sperm which need identified. To further enhance the reliability of sperm size data, author should measure the same 20 spermatozoa three, using average values in the analysis.

Validity of the findings

No comments

Additional comments

The degree of postcopulatory sexual selection, comprising variable degrees of sperm competition and cryptic female choice, is an important evolutionary force to influence sperm form and function. The authors of this paper examine intramale variation in sperm size: functional significance in a polygynous mammal. They find that high testes mass is associated with low variation in sperm size. Moreover, low variation in sperm size is strongly related to high sperm velocity and normal sperm morphology, which in turn are good predictors of male fertility in this species. Based on these results, the authors conclude that sperm size in red deer is under a strong selective force and reveal new insights into sexual selection mechanism. Overall, I enjoyed the manuscript, and think the overall data set is interesting, as is the evidence for species had distinct testicular morphological characteristics, reflecting sperm competition. The paper is generally well written: the introduction sets the scene well, methods are in general appropriate and well described, and the conclusions in the discussion are mostly well supported by the data. However, I also find that data is small in the paper. However, in the introduction for author in PeerJ, PeerJ evaluates articles based only on an objective determination of scientific and methodological soundness, not on subjective determinations of 'impact,' 'novelty' or 'interest'. Hence, I think that the paper can be published in PeerJ with a minor revision.
Authors should notice three minor comments: Firstly, authors should read trecent global reference about sperm competition within or among species in frogs. For example, recent references was in also referred sperm size in anurans by Zeng et al. (2014) and intramale variation in Mi et al. (2012) and Liao et al. (2013).
Zeng, Y., Lou, S.L., Liao W.B., Jehle, R. 2014. Evolution of sperm morphology in anurans: insights into the roles of mating system and spawning locations. BMC Evolutionary Biology, 14: 104
Liao, W.B., Mi, Z.P., Li, C.L., Wei, S.C., Wu, H. 2013. Sperm traits in relation to male amplexus position in the Omei treefrog Rhacophorus omeimontis, a species with group spawning. Herpetological Journal, 23: 23-27
Mi, Z.P., Liao, W.B.*, Jin, L., Lou, S.L., Cheng, J., Wu, H. 2012. Testes asymmetry and sperm length in Rhacophorus omeimontis. Zoological Science, 29: 368-372
Second, authors should discuss the sperm competition in frog associated with the other frog species. Third, Measurements comprised 20 sperms from each male need be done. All measurements (testes and sperm) were taken without knowledge of the species identification to prevent observer bias. The repeatability was high or low when we compared three measurements on 20 sperm which need identified. To further enhance the reliability of sperm size data, author should measure the same 20 spermatozoa three, using average values in the analysis. Finally, authors should check carefully references. See attached file.

Reviewer 2 ·

Basic reporting

The article is written in English and generally uses clear and unambiguous text. Background information on the study question and species is provided and for the most part the use of citations is appropriate, though there is a tendency for the citations to be taxonomically narrow and in some cases more recent review articles are missing. A minor review of the introductory text and supporting literature would be useful. Finally, it appears that the raw data for the study are not present in the dataset. Instead, summary data at the individual level are provided. I think it may be more appropriate to provide all (n=25) sperm trait values for each male in addition to the summaries currently supplied.

Experimental design

I have concerns regarding some of the methods used to assess sperm parameters and the low sample size. Moreover, I think some methodological details are lacking. For example, it is unclear how sperm were characterized as normal versus abnormal, nor how sperm were classified as progressively motile. Several other issues are also apparent, e.g. the use of absolute testis mass calculated as the average of the two testis, lack of control for allometry or other potentially confounding factors (e.g. mating strategy, age, etc). I have detailed these issues more fully below outlined as major comments or addressed by line number.

Validity of the findings

I am not convinced the statistical analyses used are sufficient (e.g. lack of control for confounding variables) and it seems that many results are missing from the manuscript. Most importantly, given the limitations of the sample size and the lack of control for confounds, I think the authors need to be carefully cautious in their interpretation of the results. As it stands, I think the findings are given to much weight and the speculative components of the discussion are not sufficiently identified as such. With a major revision of the manuscript, in which the results are interpreted in light of the methodological and statistical shortcomings, this paper could be suitable for publication. However, it may be preferable to conduct some new analyses incorporating confounding variables (especially mating strategy, age, etc) to improve the quality of the work presented. I have detailed some issues related to the validity of the findings more fully below, outlined as major comments or addressed by line number.

Additional comments

In this manuscript, the authors examine the relationship among intra-male variation in sperm morphology and both testis size and sperm performance (swimming speed and progressive motility). The authors suggest their results show that males with larger testis mass produce populations of highly homogenous sperm (i.e. low CV in size), and that ejaculates with low variation in sperm size have higher sperm swimming speed and a greater proportion of morphologically normal sperm. Intraspecific studies, such as this one, can significantly contribute to our understanding of patterns and processes observed across populations and species. Thus, studies such as this one are valuable, and indeed I see value in this work being published. However, I have several major concerns about the methodologies used and the presentation and interpretation of the results that must be addressed before the work should be considered acceptable for publication.

Major comments:

1. The sample size of the current study is quite low (n=17), which suggests more caution is needed in the interpretation of the results. Moreover, there is no mention of potential confounding variables (e.g. male age, male reproductive status, mating strategy, inter-population variation, seasonal variation, etc) that may influence the variables investigated.
2. I think the authors must reconsider the use of absolute testis size in their analysis. They argue that the do not control for allometry due to changes in male body condition over the course of the rut. Furthermore, they argue that species with alternate mating strategies (e.g. sneakers and harem-defenders), such as the red deer used in this study, show variation in relative testis investment. To me, this seems very problematic because male investment in testicular tissue can only be assessed in relative terms (otherwise bigger males will simply have larger testes) and because the mating tactic (i.e. sneaker v. harem-defender) of the males used in the study appears to be unknown. At the very least it is worth considering controlling for date or some other estimate of rut stage/progression and thus using relative testis size (by including both testis mass and body mass in a linear model). Similarly, it may be worth considering examining sneaker males and harem-defense males separately, or at least including this variable as a covariate in analyses. Otherwise, I don’t think you can rule out the possibility that the relationship between testis size and sperm size CV is in fact a spurious correlation generated by two clouds of data (one, perhaps sneaker males, with small testis and high sperm CV values and the other, harem-defenders, with big testis (resulting from pre-copulatory sexual selection on body size and allometry) and low sperm CV values) with no relationship between testis size and sperm CV within each of these mating strategies. Such a pattern may be evident in Figure 3C, for example, in which there seems to be a small break in testis mass between 35-35g. It would be helpful to know if the distribution of testis size is continuous or bimodal, though it is still preferable to account for body mass in analyses. Using log (mass) in analyses is also preferable. If the results did in fact show the data to be grouped by mating strategy this result would still be interesting, but this possibility strong influences the interpretation of the data.
3. Related to the above point, I think the authors should use combined testes mass in the analysis. If they can calculate an average testis mass, then data is available for both left and right testis. Thus I see no reasons to use an average value as they have done.
4. I think the authors need to more fully explain the results of the current study and consider some of the variation in their findings in the discussion. For example, the relationship between testis mass and CV in sperm morphology mentions significant negative relationships between total sperm length, head width and area and trends for negative relationships with flagellum and midpiece length. However, no mention is made of the relationship between testis mass and the CV for sperm head length, either in the text or in tables. Similarly, the negative relationships between speed (i.e. VAP and VSL) and total sperm length, flagellum length and principal piece + terminal piece are outlined in the text, but there is no mention of if/how, for example, sperm midpiece or head length are related to sperm speed (I assume there is no relationship be the exclusion of these results??). I think this is an oversight. More importantly, the authors do not attempt to explain why some traits are correlated, while others are (presumed) uncorrelated (though this is hard to be certain of due to the lacking results). Given that the authors are trying to understand the functional significance of sperm variation (CV in size), I think this is also an oversight. I would like the authors to discuss (for example) why sperm speed might be related to variation in total sperm length, but not midpiece length. Similarly, why would the % of morphologically normal sperm be related to intramale CV in sperm head measures, but not any other measures of CV in sperm size? Thus I think the authors should considerably tighten and focus the manuscript, incorporating a significant rewrite of the methods (i.e. what relationships were examined), results and the discussion, which is currently rather broad and tends towards speculation and over interpretation.
5. Linked to the above comment, is that results that VAP and VSL are significantly related to some sperm traits, but VCL is not. First, this is not explained in the discussion. More importantly, these 3 variables are typically very strongly inter-correlated and thus I think the paper suffers from over inflation of multiple testing. I recommend the authors consider a PCA of the three measures to estimate a single PC that captures all the variation of these three variables. Alternatively, a single metric can be examined if it is closely linked with competitive fertilization success in this species. That being said, in my opinion, VCL is the best descriptor of sperm motion in these types of studies because it reflects the actual sperm path and not a simpler approximation of sperm motion. I think this is important under such in vitro conditions when sperm are not expected to swim in a straight line due to the lack of a chemo-attractant and potential wall effects of the chamber slide.
6. The authors might consider some correction for multiple testing.
7. I recommend the authors are considerably more cautious in the interpretation of the results given that sperm measures were assessed at variable time points post-mortem (4-8 hours) and that this was not included as a covariate in the analysis. Though the authors provide some evidence that this delay should not influence sperm parameters, the citation they provide did not assess the same sperm traits as are used in this study. Specifically, Soler and Garde (2003) examined the sperm motility index (made up of % motile and a motility quality score), while the current study examines velocity (e.g. VSL, VAP, VCL). Thus, while the data are potentially robust, I think the authors must be open to considering potential flaws in their methodology. A considerably modified discussion of the results is therefore warranted and would go a long way in making this work acceptable for publication.

Minor comments:

Line 57 – You could also include a citation for birds (Rowe and Pruett-Jones 2011) to expand the taxonomic focus of this statement. Similarly, it may be work considering the work of Garcia-Gonzalez and Simmons (2005) on sperm viability in insects at this point.

Line 67-70 – I would consider including review papers that discuss the inconsistent relationship (at the interspecific level) between sperm morphology and swimming speed. For example, Fitzpatrick and Lupold 2014 (Molecular Human Reproduction) and Simmons and Fitzpatrick 2012 (Reproduction).

Line 74-78 – This is relative testis size and not absolute testis size, which is a very important distinction. Please add ‘relative’ to ‘testes size’ in the text.

Line 77-78 – it should be noted that there are also numerous studies showing a negative or no relationship between testis size and pre-copulatory traits. Inclusion of such studies would provide a more balanced introduction.

Line 164-166 – What constitutes a normal versus abnormal sperm? Did you do any replicate measures (e.g. 2 x 100 sperm and then average the values? Where they repeatable?)

Line 171 – How was progressive sperm motility defined? More details are necessary. For example, was it sperm at speeds over 50um/s?

Line 173 – why was VCL > 10 used as the cut-off for motile sperm?

Line 193 – how is this low, please provide some comparison to judge whether or not these values are indeed low. For example, in birds a value of 4.56 would not be considered particularly low, but more moderate – medium.

Line 225-226 – I think considerable the authors need to be more cautious in their statement that testicular mass is negatively related to sperm size variation for the reasons outlined above (lack of control for confounding variables, alternative mating tactics, lack of control for allometry and use of absolute testis mass, the use of an average testis mass and thus not a total estimate of male investment in testicular tissue).

Line 234 – Actually it is relative testes mass that has been widely used as a proxy, not testis size. This may seem a trivial distinction, but in my opinion it is an important one.

Line 246-249 – It seems to me that changes in the testicular architecture (specifically, the diameter of the seminiferous tubule lumen) is more likely to influence sperm morphology, and that such changes are more likely related to production of optimal sperm size. In contrast, there is more evidence for changes in spermatogenesis per se and testicular function impacting sperm numbers. The authors do raise the hypothesis of Mossman et al. (2013) at a later point in the discussion, which suggests that sertoli cell number may influence sperm size variability. However, this is still speculative. I think the discussion of these points needs to be considerably tightened and reduced in length.

Line 261 – 264 – the discussion of apical hooks seems off topic and could be removed.

Line 274 - The Mossman study was conducted on males undergoing treatment for infertility, thus these results need to be considered more cautiously, and I think this needs to be considered in the current manuscript also.

Line 272 – 285 – But how does the finding that the CV of some sperm traits, but not others, are related to sperm VAP/VSL and % morphologically normal sperm influence this interpretation?

Line 294- 298 – how might variation in sertoli call number be related to intra-male variation in sperm size? Moreover, if there is little variation in intramale sperm CV, is there really any potential for an association with sertoli cell number? This text is highly speculative and needs to provide more information on proposed nature of this relationship and the mechanisms underlying it.

---

## Round 0.2 · Major Revisions

Your manuscript still requires significant revisions, as per the comments of Reviewer 2.

Reviewer 1 ·

Basic reporting

I agree with publishing the manuscript.

Experimental design

It is ok.

Validity of the findings

•The submission must describe original primary research within the Scope of the journal.

Additional comments

I recommond publishing the paper.

Reviewer 2 ·

Basic reporting

No comments

Experimental design

The lack of control for body mass in this manuscript is unfortunate as it strongly limits teh interpretaion of the results. See further comments below.

Validity of the findings

Again, the inability of the authors to control for body mass or size in some for the analyses in the paper is a serious limitation of the work. The authors have done a good job in revising the manuscript based on my previous comments. However, now knowing that control for body mass/size is not possible I think this issue needs to be further addressed.

Additional comments

Overall the authors have done a very good job in reviewing the manuscript and alleviating many of my initial concerns. Nonetheless, some significant concerns remain. In particular, the fact that the authors cannot control for variation in body size/mass with respect to testes mass is unfortunate, and while I appreciate the revisions the authors have done to ensure that this issue is apparent in their work I think it prevents the paper from making firm conclusions around analyses involving testes size (i.e. relationship between testes mass and sperm CV values). The problem is that absolute testes mass does not provide information on the investment in sperm production or testes tissue by an individual. An individual with larger testes may simply be a larger individual in terms of overall size. Alternatively, in species with alternative tactics, such as the red deer in this study, it is known that sneaker males invest more heavily in testes tissue (i.e. relatively larger testes size for a given body mass) relative to harem defending males (which instead invest in pre-copulatory traits). Moreover, although a positive relationship between testes size and antler size has been observed in this species, this may simply result from variation in energy budget among individuals leasing to some males having more to invest overall and thus able to increase investment in both traits. Given this, I think the interpretation of the results concerning testes size within an evolutionary context is problematic and even more caution is needed in discussing these results. The paper is still interesting though, and with the more cautious approach to interpretation may warrant publication. Importantly, the paper does have some good findings, specifically those dealing with the relationship between sperm CV measures and sperm function, and I think this work alone is strong enough for publication (provided the low sample size is acknowledged and the authors suggest caution due to this low sample size - both of which they currently do in this revised version).

However, I see two possible approaches that the authors could take. The first, and perhaps most robust approach, that would allow the authors to draw more firm conclusions from the study, is to simply remove the parts of the study that concern intraspecific variation in testes size and how this relates to variation in sperm size (sperm CV). This would focus the paper on the relationship between sperm size variability and sperm function, which is still an interesting and important topic that needs to be addressed. This approach would also allow the paper to be more conclusive and reduce the amount of speculation, especially in the discussion.

An alternative approach would be to highlight the work on the relationship between sperm CV and sperm function and significantly reduce the importance and interpretation placed on those results dealing with variation in testes mass, placing this in a more speculative context (i.e. we find a relationship between sperm CV and sperm function and we speculate that this may be influenced by variation in testes size, and provide some data to support this suggestion. This fits with the general flow of information in the introduction anyway, and could be a good way to focus the manuscript. I have a few suggestions at different places throughout the text that may help the authors to do this. These suggestions are presented below either by section heading or by line numbers. In addition to this, the writing style could be improved at several places, and I have made some suggestions (below by line number) that may help.

Abstract – I would suggest the flow of information be changed so that the hypotheses and results concerning sperm CV and sperm function are addressed first, then the idea that intraspecific variation in testes size is introduced and followed by some preliminary evidence to support that idea. To be more transparent in the abstract, I also think the authors need to state their results in terms of absolute combined testes mass, so that the reader is well aware that it is not relative testes size being discussed.

Line 5 – delete ‘on the other hand’

Line 15-17 – I think this last sentence can be modified to focus on the findings surrounding the relationship between sperm CV and sperm function and suggest potential future research focuses, e.g. building upon the hypothesis that testes size variation may be related to producing less size variable sperm cells.

Line 46 – taxonomically widespread instead of widely spread. Also, change ‘leading’ to ‘driving’.

Line 47-48. I think the Pizzari and Parker chapter (sperm competition and sperm phenotype) is a necessary citation here. It is more appropriate than the Pitnick et al. paper, though both could be included.

Line 48 – I would strongly advise against the use of the term ‘design’, it is a very loaded term. Please use morphometry instead.

Line 53 – I think there may be a parenthesis in the wrong place as the text ‘among others’ is out of place. Please correct this sentence.

Line 54 – delete ‘every sperm counts and’ and re write the sentence as follows: ‘However, when sperm competition is intense, selection is thought to favor tight quality control’

Line 59-62 – this sentence should be reformulated to provide a stronger argument for why investigating intramale variation in sperm size is beneficial.

Line 64 – I would also include the Simmons and Fitzpatrick 2012 citation here. It has a good table summarizing the evidence for swimming speed influencing fertilization success.

Line 65 – This section is a little repetitive. I would delete the sentence beginning on line 65 and replace it with a statement that sperm size has been linked to sperm velocity in a range of inter and intra specific studies, but that results have been inconsistent. Then I would start your next sentence with ‘For example, in species with internal …..’ You will also nee to include some citations for the first sentence.

Line 68 and 70 – you don’t need ‘but see’ before either Firman and Simmons 2010 or Cramer et al. as you are not providing contrasting citations but rather support for the previous statement that results contradict.

Line 70-74 – If you include mention of interspecific studies at the beginning of this section as I have suggested above, you could delete this sentence here.


Line 78 (and throughout) – it is relative testes mass and not size that is commonly used. Though these terms seem interchangeable they are not. I suggest changing your wording here and several places throughout the manuscript.

Line 81 – I would not use the term ‘development’, this is a very specific term. Instead use something like ‘elaboration’ or ‘degree of elaboration’

Line 81 – You should also use the term pre-copulatory traits here instead of secondary sexual characters, as it is more correct and that is what these studies generally examine – the relationship between pre-copulatory and post-copulatory sexually selected traits. The definition of a secondary sexual character is one favoured by sexual selection (and not natural selection), so it could also be a trait used in post-copulatory sexual selection. I think this change will make your idea clearer. I would also suggest changing this part to be mor blanaced, saying that testes size has also been associated with variation a range of pre-copulatory, and both positive, negative and nor associations have been found.

Line 89 – Another recent intra-specific paper that could be cited here is Laskemoen et al. 2013. Behavioural Ecology 67: 301-309.

Line 93 – delete ‘the’ before calves and replace with ‘red deer’

Line 94 – delete ‘the’ before hinds

Line 95 – change to ‘), and the large numbers of aggregating females should increase the mating….’

Line 100 – delete ‘from an evolutionary perspective’

Line 101 – relative testes size was used in Malo et al. Thus add ‘relative’ before testes size

Line 101 – ‘was’ not ‘is’

Line 102 – ‘covaried’ not covary

Line 103 – the argument in this sentence is a bit weak. It is also not essential to the point of the paragraph in my opinion. The authors could consider removing it. Otherwise, as the authors themselves state in a previous place in the text, relationships between pre and post copulatory traits have been reported as positive, negative and null relationships. Thus the argument that this one study supports the idea that sperm competition is important for the evolution of male pre-copulatory traits is not well supported. If retained in the text, I would modify this sentence to something like the following: ‘, supporting the idea that elaborate sexual traits in males may signal sexual competence or sperm quality’. I would not use the term development of secondary sexual characters.

Line 105-110 – I suggest the authors place the hypothesis concerning sperm CV and sperm function first in this paragraph. This would focus this hypothesis as the main goal of the paper, which in my opinion will allow the manuscript to be more robust and improve the quality of the paper.

Line 115 – not ‘sperm fitness’ sperm don’t have fitness, this is a property of the individual. Instead say ‘by enhancing sperm competitiveness and, ultimately, male fitness’.

Line 116 – change to variation in sperm size and performance (or function)

Line 142 – please state that it is ‘absolute testes mass (i.e. the combined mass of the left and right testis)’ and not absolute testicular mass. I would also highly encourage the authors to be more transparent with the limitations of this approach at this point fin the methods and state something along the lines of the following: though the use of relative testes mass is highly preferable for studies such as these, we were unable to quantify male body mass or body size of individual stags due to the working conditions in the field. We therefore chose to use absolute testes mass (i.e. the combined mass of the left and right testis) in the current study. I think you could also add further argument here about how this approach can still be informative for the following reasons. If you can argue this point well it will strengthen the paper.

Line 164 – delete ‘a’ before red deer

Line 170 – 174 – this is a good addition to the manuscript and is very informative for the reader.

Line 180 – it seems unusual to use a makler counting chamber for this purpose. Can you perhaps state the depth or volume of the chamber so the reader can better jusge the methods.

Line 189 – 191 – I appreciate the revised methods by the authors on how progressive sperm motility was estimated. I understand this is determined by the CASA, but it would be helpful to know what STR (straightness) cut off was used for this, i.e. were cells with STR < 50 excluded? Please provide a little more information.

Line 194 – I understand that a cut off was used because dead sperm cells or debris can exhibit low levels of movement and setting a cut off avoids this problem. However, I would like the authors to provide justification for the use of 10 um/s specifically. Why not 20? Or 5? This could be as simple as referring to previous work, but it is preferable to show that dead sperm or debris move at speeds below the cut-off applied, and thus no debris or dead cells are accidently being included in analysis.

Line 200 – give the formula for CV or at least a citation for it so the reader can know the formula that was applied.

Line 205 – Testes not testicular

Line 206 – delete ‘on the other hand’

Line 207 – give the correlation coefficients (or at least the range of r values) for the intercorrelation among VCL, VAP and VSL.

Line 207 – change correlated among themselves to ‘were highly intercorrelated (all R > ?, and P < 0.???), we reduced the…..’

Line 208 – change ‘overall sperm velocity variable’ to ‘to obtain a single parameter describing sperm swimming speed’

Results section – I recommend the authors place the results concerning sperm CV and sperm function first in the results. These are the most important and robust results in the manuscript and thus I suggest these are prioritized.

Line 216 – ‘absolute testes mass’ not testicular mass.

Line 219 – change ‘the remaining’ to ‘Similarly, mean values of ….were generally low to moderate, ranging from 1.33 to 4.56’

Line 222-223 – delete ‘the’ before sperm head and before flagellum

Line 224 . which intramale CV are you referring to here, it is unclear what this sentence is actually saying.

Line 225 – delete ‘the’ before total sperm length

Line 226 – reinstate the word ‘sperm’ before head width

Line 233 – change ‘on the other hand’ to ‘Finally’

Line 236 – move this section to the beginning of the results.

Line 238- 240 – I am glad to see that the authors used a PCA to examine sperm velocity. However, I recommend revising the methods outlining the use of the PCA. The PCA would have resulted in 3 extracted PCs, but the way this is currently written it suggests only 1 PC was produced. The authors need to state the criteria they used to decide to retain only 1 PC for further analysis. For example, was this the only one with an eigenvalue > 1? Please provide additional information.

Line 257-259 (Table S1) – there are lots of correlations here, this is the point at which the authors might consider correcting for multiple testing.

Discussion section – I suggest the discussion dealing with the relationship between sperm CV and sperm function (i.e. velocity, sperm morphology) is placed first in the discussion. As noted above, this part of the paper is sound and thus focusing on this aspect of the work will improve the quality of the paper.

Line 262 – delete ‘the’ before intramale

Line 263 – change the text ‘related to’ to ‘significantly associated to’

Line 264-265 – this sentence needs to be rewritten, both in terms of clarity and also perhaps to highlight the value of this finding a little more. I would follow this with the discussion text dealing with this relationship, thus leaving the text on testes mass to the end.

Line 266 – I suggest more caution is needed here. The sentence could read something like the following: testes mass was negatively related to intramale variation in sperm size (i.e. state which specific CV values here), though we suggest this result is treated with caution as both the relative investment in testes tissue (i.e. relative testes mass) and male mating tactic (i.e. harem defender versus. sneaker) could influence this relationship.

Line 268 – the relationship has been shown for EPP and relative testes mass, not sperm competition specifically.

Line 273 – this should be something like the following: ‘our results tentatively suggest that large testes in red deer leads to sperm closer to the putative optimal size’

Line 275-284 – I would significantly reduce the length of this text as the results of the current study are not really comparable to those using relative testes mass. I would rather see the authors state something simpler like, ‘though we use absolute testes mass in this study, we found a positive relationship between testes mass and sperm number, and we speculate that this may reflect an increase in testes size in order to produce more size-uniform sperm. This idea is somewhat supported by the fact that both sperm morphology and sperm number is influenced by testicular architecture (e.g. relative proportion of spermatogenic tissue and size of the tubule lumen; Lupold et al. 2009; Rowe and Pruett-Jones 2011) and the efficiency of spermatogenesis (e.g. shorter duration of the cycle of the seminiferous epithelium; Ramm and Stockley 2010)’. I recommend the authors essentially delete all other text in this paragraph and keep this argument as simple as the suggestion above (albeit with some editing for information flow and writing style).

Line 296 – add ‘relative’ in front of testes size as Malo et al specifically looked at relative testes size.

Line 298 – 308 – this text needs to be removed or significantly shortened. This suggestion is because the text is both somewhat repetitive and is highly speculative. In line with my suggestion above to focus the manuscript on the relationship between sperm CV and sperm function, reducing this part of the discussion would significantly improve the manuscript in my opinion.

Line 308 – this text on sperm size variation should be moved up and made more of the focus as the interpretation follows more clearly and robustly from the results.

Line 328 – I would suggest this could be the start of a new paragraph (as it is currently very long). From line 328-337 – the authors have done a good job with this revised text.

Line335 – the term a ‘better manufacture sperm’ is awkward, perhaps the authors could state something like ‘ejaculates with more-uniform sized sperm (i.e. low intramale variation in length) may be more successful at achieving fertilisations under competitive mating scenarios.’

Line 342 - The idea that more uniform sized sperm is a product of efficient spermatogenesis is really only a hypothesis that I don’ think is supported by the results of the current study. I think this needs to be removed or at least considerably reworded.

Line 362 – Again, I think the statement that large testes is related to reduced variation in sperm size is somewhat speculative. The authors needs to state that it was absolute testes mass here and also tone down the imporatnance placed ont his result. The paper can indeed stand on the strength of the results concerning the relationship between CV and sperm function, and it suggest the authors revise the discussion in line with this approach.

Line 366-369 – Delete this sentence, as it is too speculative. At the very least revise it so that it is less overstated.

Legends for figures 3, 4, and 5 should be more informative. After each number (i.e. a, b, c and d) it should be clearly stated what variable that specific graph is portraying, followed by the statistical output and not just the statistics alone.

Figure 6 – I suggest the authors also include a ? on the link between testes size and sperm CV as this idea, while interesting, needs considerably more support. The strength of this paper is that it presents this idea, this is sufficient in itself and thus the results related to this idea should not be over interpreted and overstated as they currently are.

Table 2 – I think the legend for table 2 needs to be more informative. What are the P values for? What were the results for the other PCs?

---

## Round 0.3 · accepted · Accept

Authors have improved the mansucript according to all the suggestions of the reviewers.